**Data Availability Statement:** Anonymized data are available on public repository: https://github.com/TikaRamPoudel/Altitude-Sickness.

# Ascent rate and the Lake Louise scoring system: An analysis of one year of emergency ward entries for high-altitude sickness at the Mustang district hospital, Nepal

**Kapil Madi Poudel**[1,2☯], **Tika Ram Poudel**[3☯], **Neha Shah**[2,4], **Sunita Bhandari**[4], **Ramakanta Sharma**[5], **Anil Timilsina**[2], **Manab Prakash**[6☯]*

1 Institute of Medicine, Tribhuvan University, Maharajgung, Bagmati, Nepal, 2 Mustang District Hospital, Mustang, Gandaki, Nepal, 3 GoldenGate International College, Tribhuvan University, Kathmandu, Bagmati, Nepal, 4 BP Koirala Institute of Health Sciences, Dharan, Province 1, Nepal, 5 National Youth Council, Bhaktapur, Bagmati, Nepal, 6 Global Institute for Interdisciplinary Studies, Lalitpur, Bagmati, Nepal

☯ These authors contributed equally to this work.
* manab.prakash@giis.edu.np

## Abstract

More travellers are making swift ascents to higher altitudes without sufficient acclimatization or pharmaceutical prophylaxis as road connectivity develops in the Himalayan region of Nepal. Our study connects ascent rate with prevalence and severity of acute mountain sickness (AMS) among patients admitted to the emergency ward of the Mustang district hospital in Nepal. A register-based, cross-sectional study was conducted between June 2018 and June 2019 to explore associations of Lake Louise scores with ascent profile, sociodemographic characteristics, and comorbidities using chi-square test, t-test, and Bayesian logistic regression. Of 105 patients, incidence of AMS was 74%, of which 61%, 36%, and 3% were mild, moderate, and severe cases, respectively. In the Bayesian-ordered logistic model of AMS severity, ascent rate (odds ratio 3.13) and smoking (odds ratio 0.16) were significant at a 99% credible interval. Based on the model-derived counterfactual, the risk of developing moderate or severe AMS for a middle-aged, non-smoking male traveling from Pokhara to Muktinath (2978m altitude gain) in a single day is twice that of making the ascent in three days. Ascent rate was strongly associated with the likelihood of developing severe AMS among travellers with AMS symptoms visiting Mustang Hospital's Emergency Ward.

## Background

Acute mountain sickness (AMS) and high-altitude cerebral edema (HACE) can affect persons who ascend too quickly to high altitudes beyond their current acclimatization level [1]. Acclimatization ability differs dramatically, with some adjusting quickly without discomfort while others manifest AMS to such a severe degree that they appear unable to acclimatize fully and must therefore descend [2–4]. This natural acclimatization gained within the first few days is preventive against AMS and HACE. However, improvement in aerobic exercise and work

**Funding:** The author(s) received no specific funding for this work.

**Competing interests:** The authors have declared that no competing interests exist.

**Abbreviations:** AMS, Acute mountain sickness; HACE, High-altitude cerebral edema; HAPE, High-altitude pulmonary edema; LLS, Lake Louise score; LLSS, Lake Louise scoring system.

performance at high altitude requires a longer period of several weeks to months of acclimatization [2, 5–8].

AMS is characterized by headache, dizziness, fatigue, poor sleep quality, nausea, and vomiting and generally occurs within 6–12 hours after exposure to an altitude greater than 2500m above sea level [9, 10]. Normally, these symptoms are mild and self-limiting, but they can potentially be incapacitating [11] or progress to HACE, a rare but possibly fatal condition [3, 12]. Although uncommon, high-altitude pulmonary edema (HAPE) is the leading cause of altitude-illness-related death and can appear in an otherwise healthy person and progress rapidly with cough, dyspnea, and frothy sputum [13]. High-altitude illness affects 25%–85% of people who ascend to higher altitudes; risk factors include altitude reached (especially sleeping altitude), ascent rate, individual susceptibility, history of high-altitude illness, and altitude of permanent residence [13, 14]. Individual risk prediction for developing AMS is complicated, as a multitude of contributing factors such as age, sex, body mass index (BMI), ascent rate, sleeping altitude, history of AMS, history of cardiopulmonary disease, oxygen saturation ($SaO_2$), and pulse rate have been documented in the literature [15–19]. Several physiological variables including chemosensitivity and heart-rate variability have been tested, but many are not sufficiently reliable or feasible in low-resource settings [20].

Mustang, the study site, is home to several high-altitude attractions, namely the Thorong La Pass (5416m) in the world-renowned Annapurna trekking circuit, the Muktinath Temple (3800m), Lo Manthang (3840m), and the Kora La Pass (4660m). Improvements to major roadways in recent years have increased connectivity to these attractions, increased visitor flow, and shortened travel time. As a result, people are more likely to ascend quickly from lower elevations to a hypobaric hypoxia environment and, thereby, face increased risk for developing high-altitude illness during the first few days of travel [21]. An especially concerning trend is urban youth in their mid-twenties who scale altitude quickly on motorbikes traveling on newly improved roads; national media have reported multiple fatalities due to high-altitude sickness of otherwise healthy youth in the recent years.

With proper planning, the majority of AMS and HACE cases can be avoided or effectively managed [22]. Since a major reason for high-altitude emergency rescues is the development of AMS symptoms [23], any credible information on AMS will be important to travellers and health professionals taking care of them both before and after their arrival at such altitudes. This study links the ascent profile, demographic characteristics, tobacco and alcohol consumption, and comorbidities with the severity of AMS in patients visiting the Emergency Ward of the Mustang district hospital.

## Methods

### Study design

The study was designed as a retrospective register-based, cross-sectional census of patients who presented with high-altitude sickness symptoms in the Emergency Department of the Mustang district hospital in Jomsom, Nepal, from 15 June 2018 to 15 June 2019 (fiscal year 2075/76 B.S.).

### Setting

The study was conducted using patient records at Mustang district hospital. This hospital, located along the route of the Annapurna circuit at an altitude of 2743m, acts as a referral centre for primary healthcare centres and health posts receiving the majority of high-altitude sickness patients in the area, which has multiple possible routes (Fig 1).

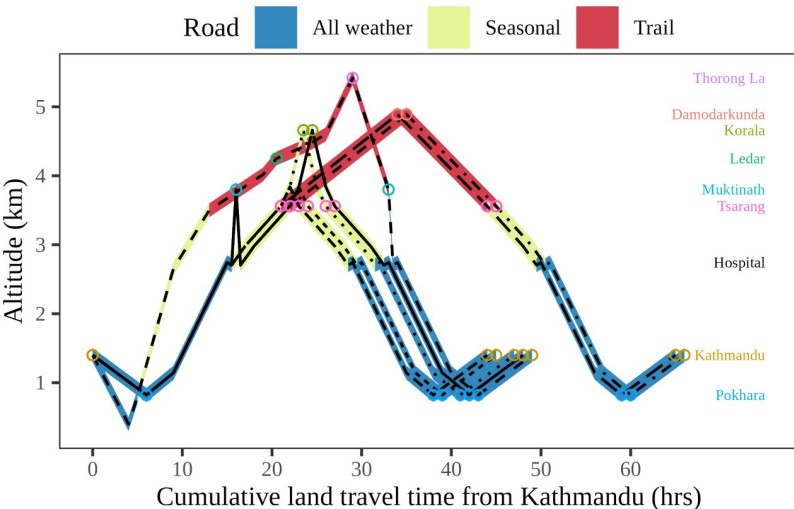

**Fig 1. Various travelling routes used by AMS patients visiting the emergency ward of Mustang district hospital.**
Each line represents a unique travelling route. Important locations in routes are plotted against their respective
elevations in the y-axis. The cumulative land-travel time on x-axis is the sum of the average land-travel time from
Kathmandu, whether by bus, jeep, or on foot, depending on the road's characteristics. All-weather roads are those that
have public bus routes. On seasonal roads, off-road four-wheelers provide transit most months of the year except
during snowfall and heavy rain. On trails, walking is the only mode of travel. Among major routes within Mustang, all
except for Damodar Kunda (4890 m) are passable with a motorized vehicle at least seasonally. Except for the route
from Manang through Thorong La, all other routes pass the Mustang district hospital during both ascent and descent.
Trekkers travelling in Thorong La route pass hospital only during the descent.

## Participants

The study population comprises travellers, pilgrims, and trekkers who reported to the Mustang
district hospital's Emergency Department with symptoms of high-altitude sickness. All com-
plete hospital records were used for the study.

## Patient and public involvement

There was no patient or public involvement in this research.

## Study tools

The Mustang district hospital used a form to assist with the clinical diagnosis and management
of high-altitude sickness. It included sociodemographic variables (age, sex, permanent
address), measurements of vital signs (blood pressure, oxygen saturation, temperature, pulse
rate), clinical signs (cyanosis), ascent characteristics, alcohol and tobacco product consump-
tion, comorbidities, medication history, Lake Louise score (LLS), and a checklist of signs and
symptoms of HAPE and HACE. It was filled by the medical officer stationed in the emergency
ward. We cleaned and transcribed the anonymized records into a digital format.

## Study variables

The outcome variable was the LLS. The Lake Louise scoring system (LLSS) 2018 is based on
the most-frequent symptoms considered important for AMS, as follows: headache, dizziness,
gastrointestinal distress (loss of appetite, nausea, or vomiting), and lassitude or fatigue [9].
Each item is scored by the subject on a scale from 0 to 3 (0 = none, 1 = mild, 2 = moderate,
3 = severe). Single item scores are added together, with the total scores ranging from a

minimum of 0 to a maximum of 12. AMS was defined as the presence of headache with at least one other symptom, and total LLS greater than or equal to 3. The LLS classifies AMS severity according to the total score; 3–5 = mild, 6–9 = moderate, and 10–12 = severe AMS. HACE is diagnosed clinically if there is a change in mental status *or* ataxia in a person with AMS or a change in mental status *and* ataxia in a person without AMS [13]. For a patient to have HAPE, he or she must have at least two of the following symptoms: dyspnea at rest, cough, weakness or decreased exercise performance, chest tightness or congestion, and at least two of the following signs: crackling sounds or wheezing in at least one lung field, central cyanosis, tachypnea, and tachycardia [13]. The response variable, AMS severity, is an ordered variable with three levels: None, Mild, and Moderate or Severe. Independent variables were ascent rate, peak altitude reached, altitude of permanent address, age, sex, smoking, alcohol intake while ascending, and history of hypertension and diabetes. Whenever country name rather than city name was recorded for permanent address, altitude of the permanent address was approximated by taking the mean elevation of the top-five largest cities in that country. We constructed ascent rate by dividing height ascended in kilometres by days spent ascending.

### Statistical analysis

Quantitative data were analysed in R 4.0.1. [24] with rstan [25] and rethinking [26] R packages using chi-square test, t-test, and Bayesian ordered logistic regression. Diffused priors were used in the Bayesian regression.

### Ethical considerations

Ethical approval was obtained from the Nepal Health Research Council (NHRC) (Ethical Review Board Protocol Registration No. 525/2021P). All methods were performed in accordance with the relevant guidelines and regulations of NHRC. Researchers had access only to anonymized medical records.

## Results

### Sociodemographic characteristics

The sample of patients (N = 105) comprised slightly more males (55.2%) than females, and mean age of all patients was 46.6 ± 20.1. Tobacco use and alcohol consumption was reported by 16.2% and 25.7% of patients, respectively (Table 1). Of 29 patients (27.6%) with at least one comorbidity, 17 (58.6%) had hypertension only, 4 (13.8%) had diabetes only, and 8 (27.6%) had both hypertension and diabetes (Table 1). Age ($\chi^2$ = 10.17, $df$ = 4, $p < 0.05$), smoking history ($\chi^2$ = 8.33, $df$ = 2, $p < 0.05$), and alcohol intake ($\chi^2$ = 18.79, $df$ = 2, $p < 0.001$) were significantly associated with the severity of AMS "S1 Table".

### AMS symptoms and vital signs in patients

About a fourth (25.7%) of 105 patients with symptoms did not meet LLS criteria to be classified as an AMS patient. Of three-quarters (74.3%) who were classified as AMS patients, 61.5% were mild, 35.9% were moderate, and 2.6% were severe. There was a single case of HACE and two cases of HAPE in our dataset. Among patients with AMS, about three-quarters (73.1%) reported gastrointestinal symptoms, 84.6% reported dizziness, and the overwhelming majority (97.4%) reported fatigue "S2 Table". Mean $SpO_2$ was lower in AMS cases compared to no AMS but the difference was not statistically significant ($t$ = 1.222, $df$ = 103, $p$ = 0.224) "S3 Table".

**Table 1. Socio-demographic characteristics of the study sample.**

| Category | Variables | Frequency (Percent of Patient) |
|---|---|---|
| **Sex** | Female | 47 (44.8%) |
| | Male | 58 (55.2%) |
| **Age** | 0–30 | 30 (28.6%) |
| | 31–60 | 46 (43.8%) |
| | ≥ 61 | 29 (27.6%) |
| **Nationality** | Nepali | 78 (74.3%) |
| | Non-Nepali | 27 (25.7%) |
| **Smoking history** | Yes | 17 (16.2%) |
| | No | 88 (83.8%) |
| **Alcohol intake** | Yes | 27 (25.7%) |
| | No | 78 (74.3%) |
| **Comorbid illness** | Yes | 29 (27.6%) |
| | No | 76 (72.4%) |
| **Medication history** | Yes | 20 (19.0%) |
| | No | 85 (81%) |

Alcohol intake: drinking alcohol during the travel duration. Smoking history: patient is an active smoker. Medication history: patient is taking any medication for comorbidities or illness.

## Model of AMS severity

Table 2 presents the parameter estimates (log odds) of the Bayesian-ordered logistic model of AMS severity. We added ascent characteristics and altitude of permanent address, demographic characteristics, smoking and alcohol intake, and comorbid conditions as explanatory variables in stepwise fashion. Adding variables increased the coefficient of ascent rate (odds

**Table 2. Bayesian ordered logistic model of AMS severity.**

| | | Logistic model of AMS severity | | | |
|---|---|---|---|---|---|
| | | **(1)** | **(2)** | **(3)** | **(4)** |
| Cut point 1 | | -0.93 (1.98) | -0.89 (2.01) | -1.01 (2.03) | -0.97 (2.03) |
| Cut point 2 | | 1.16 (1.98) | 1.32 (2.01) | 1.39 (2.03) | 1.48 (2.03) |
| Ascent rate (km/day) | | 0.54* (0.29) | 0.61* (0.34) | 0.90** (0.37) | 1.14*** (0.40) |
| Peak altitude reached (Log) | | 0.01 (0.27) | 0.03 (0.28) | 0.00 (0.28) | -0.02 (0.28) |
| Permanent address altitude (Log) | | -0.09 (0.14) | -0.13 (0.14) | -0.10 (0.15) | -0.08 (0.15) |
| Age (0–30 base category) | 31-60yr | | 0.47 (0.48) | 0.44 (0.49) | 0.68 (0.52) |
| | 61-85yr | | -0.87 (0.55) | -0.96* (0.56) | -0.79 (0.59) |
| Female | | | 0.34 (0.40) | 0.16 (0.44) | 0.09 (0.45) |
| Smoking | | | | -1.79*** (0.59) | -1.82*** (0.6) |
| Alcohol intake while ascending | | | | 0.48 (0.47) | 0.4 (0.47) |
| Presence of hypertension or diabetes | | | | | -0.95* (0.53) |
| Observations | | 105 | 105 | 105 | 105 |
| WAIC | | 227.06 (7.64) | 225.56 (8.88) | 219.35 (11.20) | 218.59 (12.24) |
| PSIS | | 227.08 (7.68) | 225.60 (8.92) | 219.45 (11.26) | 218.72 (12.30) |

Coefficients are log odds; Sampled posterior standard error in parentheses;

***, **, and * imply population effects that are different from zero in 99, 95 and 90 credible intervals respectively.

WAIC is Watanabe–Akaike information criterion. PSIS is Pareto smoothed importance sampling statistic.

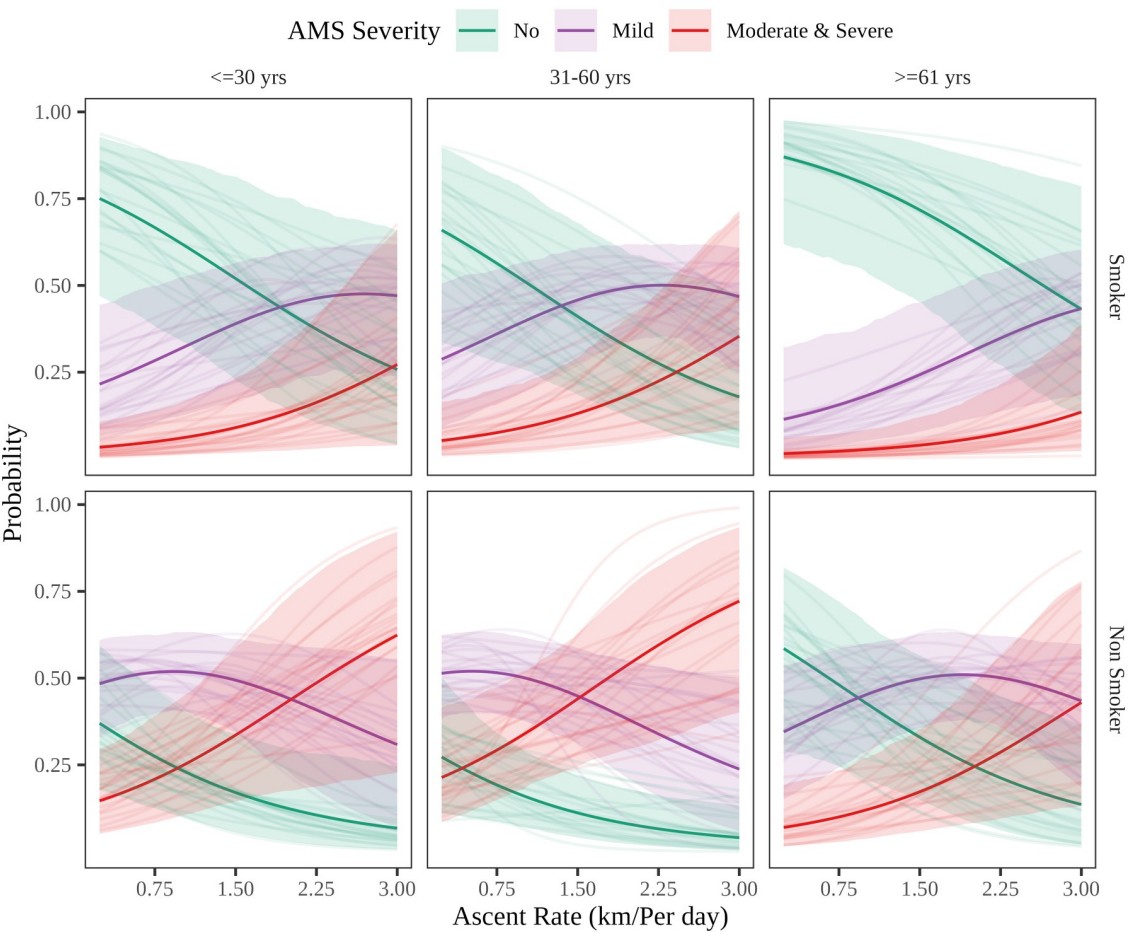

**Fig 2. The counterfactual probability plot for AMS severity with ascent rate, age, and smoking status derived from model 4 of Table 2.** The baseline is represented by a male with no comorbidities, with a permanent address at an altitude of 542m (median in the dataset), reaching the peak altitude of 3800m (altitude of Muktinath temple) without drinking alcohol during the ascent. Bold lines are posterior means. Shaded areas are posterior 95% credible intervals, and thin lines are sampled fitted lines from the posterior. For a middle-aged (31–60 years), non-smoker male travelling from Pokhara to Muktinath in a single day, the odds of developing moderate or severe AMS increase twofold (2.069) compared to those of a three-day journey.

ratio 3.13) and made it significant at a higher credible interval (i.e. 99%). Age was significant at a 90% credible interval in the third model, but the addition of the comorbidities variable removed this association in the final model. In the fourth model, smoking (odds ratio 0.16) and presence of diabetes or hypertension are negatively associated with AMS severity at 99% and 90% credible intervals, respectively. The positive association between ascent rate and AMS severity is reflected for every subcategory in the counterfactual plot (Fig 2).

## Discussion

The study shows that the risk of developing AMS for a given risk profile can be substantially reduced in severity by decreasing the ascent rate. Additionally, AMS was associated negatively with smoking behaviour and the presence of hypertension and/or diabetes. Age group was significant in the absence of the comorbidity variable, while sex and alcohol intake were not statistically significant.

A slower ascent rate allows time to acclimatize to the high altitude, thereby decreasing the intensity of AMS [27–29]. In a previous study, male volunteers who travelled to 3500m within

two days were less affected by AMS than males randomized to a 2-hour airlift to the same altitude [27]. Similar results have been reported elsewhere [30–33]. The planned road development in the study area can potentially reduce travel time from Beni to Muktinath, a distance of 100km, to 3 hours. Ascending 2901m of altitude within a couple of hours increases the likelihood of developing severe AMS; our results support this claim.

Previously, it was hypothesized that smokers would suffer more from AMS with problems acclimatizing to high altitude because of aggravated hypoxaemia from carboxyhaemoglobin, decreased oxygen uptake, and diminished peripheral oxygen extraction [34]. However, multiple studies report a lower likelihood of developing AMS among smokers than non-smokers [35–37], which is also supported by our results. It is speculated that smoking impairs endothelial function, decreasing nitric oxide (NO) formation and increasing NO degradation, protecting somewhat from headache and gastrointestinal upset [38]. However, the overall deleterious impact of smoking on health, exercise capacity and frostbite susceptibility [39, 40] mean it cannot be recommended for AMS prevention.

In the literature [41, 42], a negative correlation has been identified between age and AMS even when adjusting for ascent rate. A similar correlation is depicted by our third model, but the addition of the comorbidity variable removed this association in the final model; the most likely explanation is the collinear nature of comorbidity and age. In counterfactual plots, with an older age group, the odds of more severe AMS decrease, but the confidence intervals are too wide for the results to be statistically significant; a larger dataset might have clarified this issue. The reason for the age–AMS negative correlation might be age-related physiological changes like differences in intracranial and intraspinal cerebrospinal fluid capacity due to decreased brain volume [43] and changes in ventilatory and cardiac responses to hypoxia [44]. We do not find any association between sex and AMS severity; in fact, this association has been a mixed bag in the literature. Some papers [10, 45] report a higher incidence of AMS in females than males, but in others [42, 46], there was no association with sex. The general hypothesis is that alcohol intake may intensify AMS as it dehydrates the body, depresses breathing, and decreases ventilation [47], but we do not find support for this in our dataset. Presence of comorbid illness was negatively associated with AMS severity at 90% credible interval in the final model. We cannot be sure if this effect is because of comorbidity or old age.

Multiple studies have used body mass index (BMI) as a study variable. However, we could not use BMI as one of the explanatory variables since we only had access to the AMS-specific form that did not contain information to calculate BMI. This omission potentially decreases the explanatory power of models used in this study since some studies report BMI as one of the risk-factor for altitude sickness [18, 48, 49]. In contrast, other report non-significance of BMI [50]. Other limitations of the study arise from sampling-related issues. The study sample comprises the sample of high-altitude sickness patients conditional on visiting the district hospital and, thus, does not represent a general distribution. The second source of potential sampling bias arises from the hospital's location. If patients showed signs of HAPE or HACE when they were far away from the hospital, they were usually airlifted to tertiary medical centres. If they were closer, they would first have been taken to the Mustang district hospital for early management before being airlifted. Additionally, the Annapurna circuit pruned AMS patients in two ways: first, Thorong La Pass sieved less physically capable travellers from reaching Mustang, and second, travellers suffering from severe AMS were airlifted directly to tertiary medical centres.

## Conclusion

Ascent rate has a significant influence on the development of greater AMS intensity. A middle-aged male non-smoker travelling from Pokhara to Muktinath in a single day has a twofold

increase in the risk of developing moderate or severe AMS compared to a three-day trip. In light of recent and potential roadway development, AMS patients at the Mustang district hospital are likely to increase unless appropriate preventive interventions are implemented.

## Supporting information

**S1 Table. Socio-demographic characteristics, comorbidities and AMS severity distribution.**
(PDF)

**S2 Table. Distribution of symptoms' intensity and symptoms' presence in patients with AMS.**
(PDF)

**S3 Table. Variation in vital signs with AMS severity.**
(PDF)

**S1 Checklist. STROBE statement—Checklist of items that should be included in reports of *cross-sectional studies*.**
(DOCX)

## Acknowledgments

The authors are grateful to the reviewers, Keisuke Hatano and Bengt Kayser, and an anonymous editor whose substantiative comments helped to polish the manuscript. We are thankful to NHRC, Mustang district hospital, Dr. Santosh Dev, and Dr. Ram Sharma Subedi for ethical clearance, study design, and administration. Feedbacks from Dr. Buddha Basnyat, Dr. Sanjiv Bhandari, Dr. Santosh Baniya, Dev Ram Sunuwar, Prakash Chandra Aryal, Dr. Resham Thapa-Parajuli, Tilak Kshetri, and Sanjeet Singh Thapa were helpful in polishing the final manuscript. Error and omissions, if any, are ours.

## Author Contributions

**Conceptualization:** Kapil Madi Poudel, Tika Ram Poudel, Neha Shah, Sunita Bhandari, Ramakanta Sharma, Manab Prakash.

**Data curation:** Kapil Madi Poudel, Tika Ram Poudel, Neha Shah, Anil Timilsina, Manab Prakash.

**Formal analysis:** Kapil Madi Poudel, Tika Ram Poudel, Manab Prakash.

**Investigation:** Kapil Madi Poudel, Tika Ram Poudel, Neha Shah, Manab Prakash.

**Methodology:** Kapil Madi Poudel, Tika Ram Poudel, Manab Prakash.

**Project administration:** Kapil Madi Poudel, Tika Ram Poudel, Neha Shah.

**Resources:** Kapil Madi Poudel, Tika Ram Poudel.

**Software:** Kapil Madi Poudel, Tika Ram Poudel, Manab Prakash.

**Supervision:** Tika Ram Poudel, Sunita Bhandari, Ramakanta Sharma.

**Validation:** Kapil Madi Poudel, Tika Ram Poudel, Manab Prakash.

**Visualization:** Tika Ram Poudel, Manab Prakash.

**Writing – original draft:** Kapil Madi Poudel, Tika Ram Poudel, Manab Prakash.

**Writing – review & editing:** Kapil Madi Poudel, Tika Ram Poudel, Manab Prakash.

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
