## [Decision Letter · Decision Letter 0]

7 Sep 2022

PONE-D-22-23029Ascent rate and the Lake Louise scoring system: An analysis of one year of emergency ward entries of high-altitude sickness at the Mustang District Hospital, NepalPLOS ONE

Dear Dr.Manab Prakash ,

Thank you for submitting your manuscript to PLOS ONE. After careful consideration, we feel that it has merit but does not fully meet PLOS ONE’s publication criteria as it currently stands. Therefore, we invite you to submit a revised version of the manuscript that addresses the points raised during the review process.

ACADEMIC EDITOR: Thank you very much for the manuscript. As two reviewers suggest "major revision", please carefully revise the manuscript and resubmit it. 

We look forward to receiving your revised manuscript.

Kind regards,

Ayataka Fujimoto

Academic Editor

PLOS ONE

Journal Requirements:

4. Ethics statement appears in the Methods section of the manuscript AND at the end of the manuscript:

Your ethics statement should only appear in the Methods section of your manuscript. If your ethics statement is written in any section besides the Methods, please delete it from any other section. 

Additional Editor Comments:

Please carefully see the comments form the reviewers and revise the manuscript in accordance with the comments.

Reviewers' comments:

Reviewer's Responses to Questions

**Comments to the Author**

1. Is the manuscript technically sound, and do the data support the conclusions?

Reviewer #1: Yes

Reviewer #2: Yes

2. Has the statistical analysis been performed appropriately and rigorously? 

Reviewer #1: I Don't Know

Reviewer #2: Yes

3. Have the authors made all data underlying the findings in their manuscript fully available?

Reviewer #1: Yes

Reviewer #2: Yes

4. Is the manuscript presented in an intelligible fashion and written in standard English?

Reviewer #1: Yes

Reviewer #2: Yes

5. Review Comments to the Author

Reviewer #1: This is an important study reporting the risk factors of AMS. Please correct the following points.

1. You mentioned odds ratio in the Abstract, but not in the Result. It is better to describe odds ratio in the Result.

2. In the Methods, you did not mention the detail about the Bayesian-ordered logistic model. Please describe more about the study design or analysis. The sentence “The responsive variable, AMS severity, is an ordered variable with three levels…” in 3.3 is usually mentioned in the Methods.

3. Table 1 does not show data about 95%CI. It is better to remove “[95%CI]” in Table 1.

4. In Figure 1, what does each line represent? Is it travelling routes? Are there seven routes? And please clarify what Thorangla, Damodarkunda Korala, Ladar, Muktinath Tsarang, Hospital, Kathmandu, and Pokliara represents.

5. In Table 2, what does every data represent? (log odds?) What are “Cut point 1”, “Cut point 2”, “WAIC”, and “PSIS”? Could you please clarify these?

6. In the first of the Result, could you please describe how many candidates were included in this study.

7. In 3.1 Sociodemographic characteristics, what do Ｘ2 and df represent? Please clarify these words or remove them.

8. P-value of smoking history was <0.01 in 3.1 Sociodemographic characteristics, but 0.015 in SI Table1. Which one is correct?

9. Please change “Km” to “km” in Table 2, Figure 1 and 2.

10. Please describe the limitations of this study in the Discussion. I think the content you mentioned in 2.7 Bias is a limitation.

Reviewer #2: General

This manuscript reports the results of a retrospective study analyzing hospital admission for emergencies for the presence of altitude illnessess (AMS, HACE and HAPE) in a remote high altitude region in Nepal. The authors are commended for their work which provides a welcome addition to the literature in this field.

Major

What is the reason that the authors did not include BMI in their analysis? Given they had full access to the hospital records I presume that stature and body mass were available. If they have collected those data I strongly suggest to include BMI (or a similar compound variable) in their analyses and report the results.

Minor

L2, perhaps better to write ‘entries for high-altitude sickness’

L24-25, I suggest: ‘Our study connects ascent rates with prevalence and severity of acute mountain sickness (AMS) among patients admitted to the emergency ward of the Mustang district hospital in Nepal.’

L29, in the abstract I suggest to omit the decimals of the percentages

L40, I suggest to add ‘can’ before ‘affect’

L43-44, I suggest: ‘to such a severe degree that they appear unable to acclimatize fully and must therefore descend.’

L46, ‘several weeks to months’

L51, ‘a rare but possibly fatal condition’

L56, as written the permanent residence is a bit confusing. I understand what the authors try to convey, but it is a bit unclear, please reformulate.

L59, use subscript in the molecule di-oxygen

L91, ‘the emergency ward of the Mustang’

L105, ‘The Mustang district hospital uses’

L106, ‘included’

L111, perhaps add that the data were anonymized.

L120, HACE without article.

L149, add anonymization

L167, ‘was lower’

L196, add ‘negatively’

L198, ‘while sex and’

L221, delete ‘would’

L226, ‘been a mixed bag’

L237, ‘at the Mustang district hospital’

References, some missing references to estimations of altitude illness in that region, especially around the Thorong La

Table 1, missing 95% CI, needs legend explaining what Smoking history, Alcohol intake and Medication history stand for

Table 2, spell out WAIC and PSIS in the legend

Figure 1, I believe it is Thorong La?

6. PLOS authors have the option to publish the peer review history of their article (what does this mean?). If published, this will include your full peer review and any attached files.

Reviewer #1: **Yes: **Keisuke Hatano

Reviewer #2: **Yes: **Bengt Kayser

---

## [Author Response · Author response to Decision Letter 0]

24 Sep 2022

Thank you for very important comments. We have tried to address them best of our abilities. Further, we have attached the separate doc file named response to reviewer with the submission that goes point by point to reviewers/editors comments.

---

## [Decision Letter · Decision Letter 1]

7 Oct 2022

PONE-D-22-23029R1Ascent rate and the Lake Louise scoring system: An analysis of one year of emergency ward entries for high-altitude sickness at the Mustang district hospital, NepalPLOS ONE

Dear Dr. Prakash,

Thank you for submitting your manuscript to PLOS ONE. After careful consideration, we feel that it has merit but does not fully meet PLOS ONE’s publication criteria as it currently stands. Therefore, we invite you to submit a revised version of the manuscript that addresses the points raised during the review process.

ACADEMIC EDITOR: Please insert comments here and delete this placeholder text when finished. Be sure to:Please respond to the reviewer's comments which are minor revision as a round2.  Please ensure that your decision is justified on PLOS ONE’s publication criteria and not, for example, on novelty or perceived impact.

We look forward to receiving your revised manuscript.

Kind regards,

Ayataka Fujimoto

Academic Editor

PLOS ONE

Journal Requirements:

Additional Editor Comments (if provided):

Please respond to the comment form the reviewer who suggests "minor revision". It seems that there are some errors in terms of a finger space, especially in the results area. For example, L155, %of etc.

Reviewers' comments:

Reviewer's Responses to Questions

**Comments to the Author**

1. If the authors have adequately addressed your comments raised in a previous round of review and you feel that this manuscript is now acceptable for publication, you may indicate that here to bypass the “Comments to the Author” section, enter your conflict of interest statement in the “Confidential to Editor” section, and submit your "Accept" recommendation.

Reviewer #1: (No Response)

Reviewer #2: All comments have been addressed

2. Is the manuscript technically sound, and do the data support the conclusions?

Reviewer #1: (No Response)

Reviewer #2: Yes

3. Has the statistical analysis been performed appropriately and rigorously? 

Reviewer #1: (No Response)

Reviewer #2: Yes

4. Have the authors made all data underlying the findings in their manuscript fully available?

Reviewer #1: (No Response)

Reviewer #2: Yes

5. Is the manuscript presented in an intelligible fashion and written in standard English?

Reviewer #1: (No Response)

Reviewer #2: Yes

6. Review Comments to the Author

Reviewer #1: 1. Thank you for modifying Figure 1 capture. However, Figure 1 is still a little difficult to understand. For example, it is not clear if people pass through the Hospital when they are climbing or when they are descending (or both?). It is also unclear whether people pass through Ledar in all routes. How about using marks or colors such as ☆, ＊, ■, …etc. for Thorong La, Damodarkunda Korala, Ledar, Muktinath Tsarang, Hospital, Kathmandu, and Pokhara? I think it will be clear where to pass at which point in the figure. Moreover, the overlap between the line of routes and the word “Kathmandu and Pokhara” is not good and should be modified.

2. Thank you for explanation of Table 2. You described “AMS severity” in the first row of Table 2. However, I think “logistic model” or “logistic model of AMS severity” is better and more understandable because (1), (2), (3), and (4) in the second row shows not AMS severity but logistic model. Please consider correcting if my interpretation is correct.

3. In Table 2, I understood that Cut point 1 separated between None and Mild and that Cut point 2 separated between Mild and Moderate or Severe. How about describing like “Cut point 1 (None vs Mild, Moderate, or Severe)” and “Cut point 2 (None or Mild vs Moderate or Severe )” if my interpretation is correct.

4. You answered that the odds ratio is calculated as ratio of P(1day|X)/ P(3day|X) in Fiugre 2 and that you have explained textually about the odds ratio. However, I think Figure 2 doesn’t show that the odds ratios of ascent rate and smoking are 3.13 and 0.16, respectively, which are described in the Abstract. Does Figure 2 reveal the odds ratio of smoking? Moreover, the Figure 2 caption did not explain about odds ratios of ascent rate and smoking. The result described in the Abstract should be also written in the manuscript, Table, Figure, or Figure caption. Please reconsider.

Reviewer #2: I thank the authors for addressing all of my comments and suggestions. I have no more additional points to make.

7. PLOS authors have the option to publish the peer review history of their article (what does this mean?). If published, this will include your full peer review and any attached files.

Reviewer #1: No

Reviewer #2: **Yes: **Bengt Kayser

---

## [Author Response · Author response to Decision Letter 1]

14 Oct 2022

We have uploaded the response to reviewer file addressing all comments.

---

## [Decision Letter · Decision Letter 2]

17 Oct 2022

Ascent rate and the Lake Louise scoring system: An analysis of one year of emergency ward entries for high-altitude sickness at the Mustang district hospital, Nepal

PONE-D-22-23029R2

Dear Dr. Manab Prakash,

We’re pleased to inform you that your manuscript has been judged scientifically suitable for publication and will be formally accepted for publication once it meets all outstanding technical requirements.

Kind regards,

Ayataka Fujimoto

Academic Editor

PLOS ONE

Additional Editor Comments (optional):

Dear authors,

All the reviewers accept the latest version.

Thank you very much for the revisions.

This paper will be beneficial to all readers.

Reviewers' comments:

Reviewer's Responses to Questions

**Comments to the Author**

1. If the authors have adequately addressed your comments raised in a previous round of review and you feel that this manuscript is now acceptable for publication, you may indicate that here to bypass the “Comments to the Author” section, enter your conflict of interest statement in the “Confidential to Editor” section, and submit your "Accept" recommendation.

Reviewer #1: All comments have been addressed

2. Is the manuscript technically sound, and do the data support the conclusions?

Reviewer #1: Yes

3. Has the statistical analysis been performed appropriately and rigorously? 

Reviewer #1: Yes

4. Have the authors made all data underlying the findings in their manuscript fully available?

Reviewer #1: Yes

5. Is the manuscript presented in an intelligible fashion and written in standard English?

Reviewer #1: Yes

6. Review Comments to the Author

Reviewer #1: (No Response)

7. PLOS authors have the option to publish the peer review history of their article (what does this mean?). If published, this will include your full peer review and any attached files.

Reviewer #1: No

---

## [Editor Report · Acceptance letter]

19 Oct 2022

PONE-D-22-23029R2 

Ascent rate and the Lake Louise scoring system: An analysis of one year of emergency ward entries for high-altitude sickness at the Mustang district hospital, Nepal 

Dear Dr. Prakash:

I'm pleased to inform you that your manuscript has been deemed suitable for publication in PLOS ONE. Congratulations! Your manuscript is now with our production department. 

Kind regards, 

on behalf of

Dr. Ayataka Fujimoto 

Academic Editor

PLOS ONE